# New Trends in the Detection of Gynecological Precancerous Lesions and Early-Stage Cancers

**DOI:** 10.3390/cancers13246339

**Published:** 2021-12-17

**Authors:** Jitka Holcakova, Martin Bartosik, Milan Anton, Lubos Minar, Jitka Hausnerova, Marketa Bednarikova, Vit Weinberger, Roman Hrstka

**Affiliations:** 1Research Centre for Applied Molecular Oncology, Masaryk Memorial Cancer Institute, 656 53 Brno, Czech Republic; holcakova@mou.cz (J.H.); martin.bartosik@mou.cz (M.B.); 2Department of Obstetrics and Gynecology, Masaryk University and University Hospital, 625 00 Brno, Czech Republic; anton.milan@fnbrno.cz (M.A.); minar.lubos@fnbrno.cz (L.M.); 3Department of Pathology, Masaryk University and University Hospital, 625 00 Brno, Czech Republic; hausnerova.jitka@fnbrno.cz; 4Department of Internal Medicine, Hematology and Oncology, Masaryk University and University Hospital, 625 00 Brno, Czech Republic; bednarikova.marketa@fnbrno.cz

**Keywords:** cervical cancer, endometrial cancer, ovarian cancer, precancer, liquid biopsy

## Abstract

**Simple Summary:**

This review deals with the prospects of early diagnostics in cervical, endometrial, and ovarian cancers. Progress in cancer research has enabled the rapid development of cancer diagnostics, treatments, and preventions. Indeed, the current situation in genital tract tumors reflects the trend in contemporary oncology that emphasizes cancer prevention and early diagnostics. Cervical cancer screening using cytological examination has a long tradition, is highly effective, and is now being complemented with HPV triage, allowing early diagnostics even in precancerous stages due to the relative ease of accessibility of the uterine cervix. Moreover, vaccination against HPV, which is major cause of cervical cancer, is now recommended for both girls and boys at an early age. In contrast, endometrial and, particularly, ovarian cancers are poorly accessible by sampling, and neither prevention nor screening methods readily exist. Thus, other options leading to their early diagnostics are discussed, in which circulating biomarkers play key roles.

**Abstract:**

The prevention and early diagnostics of precancerous stages are key aspects of contemporary oncology. In cervical cancer, well-organized screening and vaccination programs, especially in developed countries, are responsible for the dramatic decline of invasive cancer incidence and mortality. Cytological screening has a long and successful history, and the ongoing implementation of HPV triage with increased sensitivity can further decrease mortality. On the other hand, endometrial and ovarian cancers are characterized by a poor accessibility to specimen collection, which represents a major complication for early diagnostics. Therefore, despite relatively promising data from evaluating the combined effects of genetic variants, population screening does not exist, and the implementation of new biomarkers is, thus, necessary. The introduction of various circulating biomarkers is of potential interest due to the considerable heterogeneity of cancer, as highlighted in this review, which focuses exclusively on the most common tumors of the genital tract, namely, cervical, endometrial, and ovarian cancers. However, it is clearly shown that these malignancies represent different entities that evolve in different ways, and it is therefore necessary to use different methods for their diagnosis and treatment.

## 1. Introduction

Current diagnostics for genital tract tumors, i.e., cervical, endometrial, and ovarian tumors, reflect the situation in contemporary oncology. On the one hand, there is an existing screening program for cervical cancer (CC) that is well-organized in developed countries and where cytological screening, in combination with colposcopy and HPV triage, is responsible for a dramatic decline in invasive cancer incidence and mortality [1]. On the other hand, endometrial and ovarian cancers are diagnosed only when the first symptoms appear, i.e., there is no screening available. In endometrial cancer (EC), ultrasound examination is proposed in patients with known risk factors, but further verification by endometrial biopsy is necessary. In ovarian cancer (especially epithelial ovarian cancer, EOC), it is still not possible to shift diagnostics from the advanced to the early stages of disease [2,3,4]. The current practice, which combines an ultrasound examination with detection of CA-125, is not sensitive and reliable enough to detect early stages. Therefore, the implementation of new biomarkers that would allow for non-invasive early detection is highly desirable.

A number of current studies focus on the use of biomarkers obtained using liquid biopsies from various sources in the human body, such as uterine and peritoneal lavages, cervicovaginal fluids, saliva, urine, and blood [5]. Liquid biopsies (LB) provide a great opportunity to overcome many of the limitations of the traditional biopsy in the detection, analysis, and monitoring of cancer in various bodily fluids, instead of cancer tissue fragments (Figure 1). In addition to providing accurate information about the presence and specific molecular features of cancer, LB can reflect the body’s response to the chosen treatment regimen, detect early recurrence, and identify possible causes of treatment resistance. In addition, they can help detect early tumors that are not visible by conventional imaging [6].

In this review, we present the current screening options for gynecological precancerous lesions and describe novel liquid-based biomarkers that may be useful in the early cancer and precancer diagnostics of gynecological malignancies.

## 2. Overview of Circulating Biomarkers

An LB is the collection and analysis of non-solid biological tissues, especially blood, but also urine, sputum, uterine lavage, cerebrospinal fluid, saliva, stool, and pleural fluid (Figure 2). The main components of an LB, in relation to cancer, are circulating tumor cells (CTC), circulating tumor DNA (ctDNA), circulating cell-free RNA (microRNA and long non-coding RNA), and circulating extracellular vesicles, especially exosomes (EX), which are subjected to various analyses. For the prevention of gynecological malignancies in clinical practice in addition to blood sampling, it is also possible to use standard samples taken by minimally invasive methods such as Pap smears, uterine or vaginal swabs, etc. Table 1 lists examples of studies that tested these samples to detect ovarian, endometrial, and cervical tumors. A summary of the most commonly used biological components and molecular methods for the research and diagnosis of individual diseases is shown in Figure 1.

### 2.1. Circulating Tumor Cells

Circulating tumor cells (CTCs) are tumor cells that enter the bloodstream from primary tumors, relapses, or metastases [22]. CTCs are thought to cause metastases [23] and have been reported in various cancers, including breast, lung, liver and prostate cancer [24]. CTCs are extremely rare (estimated to be one CTC per 10^9^ normal blood cells in the circulation of patients with advanced cancer). The use of CTCs is therefore limited by the availability of technologies for their isolation in sufficient quantities and under conditions that are compatible with subsequent analyses [25]. CTCs are commonly enriched in two ways: cell surface marker-dependent and marker-independent approaches. The epithelial cell adhesion molecule (EpCAM), a marker on the surface of epithelial cells that is absent in normal leukocytes, is the most commonly used marker in the first approach [25]. Marker-independent CTC isolation utilizes the biophysical properties of CTCs, such as their size, deformability, or dielectric susceptibility. The obtained CTCs can be used for a variety of cytological and molecular analyses (genomic and transcriptomic) [26]. Viable CTCs can serve in functional studies, such as in the creation of xenograft models in immunocompromised mice [27], the establishment of cell lines [28], and the formation of spheroids using 3D cultures [29,30].

### 2.2. Cell-Free DNA and Circulating Tumor DNA

Cell-free DNA (cfDNA) refers to all non-encapsulated DNA in body fluids. If the cfDNA originates from tumor cells, it is called circulating tumor DNA (ctDNA). CfDNA is made up of fragments of nucleic acids (approximately 170 bp) that enter the bloodstream during cell apoptosis or necrosis. The half-lives of these fragments are approximately two hours, which allows for the dynamic monitoring of cancer development. The amount of cfDNA can serve as a biomarker of cancer, but only an analysis of tumor-derived ctDNA provides the necessary sensitivity and specificity. CtDNA is used to detect various genetic and epigenetic changes, such as chromosomal abnormalities, the specific loss of heterozygosity (LOH), somatic and germline gene mutations, and aberrant DNA methylation that may correlate with cancer diagnoses, prognoses, or responses to treatment [31].

### 2.3. Circulating Cell-Free RNA

MicroRNAs (miRNAs) are endogenous, small non-coding RNAs (21–25 bps) that function as oncogenes or tumor suppressor genes under certain conditions [32]. They are the most abundant RNA molecules circulating in the blood and can be transmitted in exosomes, apoptotic bodies, or protein–miRNA complexes. MiRNA shows a high stability in biological samples (probably due to its protection in exosomes and/or protein complexes). Body fluids can also contain long non-coding RNAs (lncRNAs) that represent RNA molecules longer than 200 nucleotides with the ability to modify the activity of various genes (by the activation and enhancement of gene transcription, or by the post-transcriptional regulation of mRNA splicing and translation). The amounts and profiles of miRNAs and lncRNAs vary in the body fluids of cancer patients and healthy subjects and can be used as potential cancer biomarkers [33,34].

### 2.4. Circulating Extracellular Vesicles

Exosomes (EX) are small extracellular vesicles with a diameter of 40–100 nm that are actively secreted by most eukaryotic cells into the surrounding body fluids. These vesicles mediate communication between cells by transporting signaling molecules such as lipids, proteins, and nucleic acids, and are involved in various physiological and pathological processes. Exosomes released from tumor cells often contain oncogenic molecules and their molecular cargo differs from exosomes delivered by normal cells, so they can be used as non-invasive biomarkers for the early diagnosis and prognosis of most cancers. Their large amount and stability in different body fluids allows for their non-invasive acquisition and they are therefore considered a suitable diagnostic biomarker for distinguishing different stages of cancer [35], showing a higher sensitivity and specificity compared to other biomarkers derived from body fluids [36]. Since different types of tumors are characterized by tumor-specific miRNAs or proteins, the exosomal cargo released from their cells reflects the degree of tumor progression and the treatment outcome [37,38]. However, it is essential to select a suitable isolation protocol according to the type and volume of the initial sample, and with respect to the subsequent analysis, to obtain optimal results [39,40].

## 3. Cervical Precancerous Lesions

### 3.1. Current Diagnostics of Cervical (Pre)Cancer

In 2020, cervical cancer (CC) was the fourth most common cancer in women worldwide, both in incidence (with >600,000 new cases) and mortality (>340,000 deaths) (https://gco.iarc.fr/, accessed on 22 June 2021). It is well-known that an infection with high-risk types of human papillomavirus (HR HPV) is a major etiological factor of CC, responsible for the majority of cases. The relatively easy accessibility of the cervix enables clinicians to search for precancerous lesions directly in samples collected with a cervical brush. For a long time, the cytology-based Pap test (or Pap smear) was the only method of screening, which undoubtedly helped to reduce overall incidence and mortality, especially in the Western world. However, its high rate of sampling errors, presence of obscuring material (such as blood or mucus), and interpretation mistakes led to the development of its more practical, faster, less obscured, and more accurate alternative, called liquid-based cytology (LBC) [41]. Compared to conventional cytology, LBC is shown to have a higher sensitivity for detecting low-grade squamous intraepithelial lesions (LSILs), but not for high-grade squamous intraepithelial lesions (HSILs) [42]. On the other hand, LBC offers another practical advantage over conventional cytology, i.e., the sample in the liquid medium can be used in parallel for HPV testing (described below) without the need for further sample collection.

Despite the success of cytology-based methods, they offer only moderate sensitivities for the diagnosis of HSILs. A large clinical trial, ATHENA [43], showed that molecular-based HPV DNA testing was a more sensitive alternative, demonstrating that one in four women who were HPV 16-positive would have cervical disease within three years and that nearly one in seven with normal Pap cytology who were HPV 16-positive had a high-grade cervical lesion that was missed by cytology. Based on those results, in 2014, the FDA approved the use of the Roche Cobas^®^ HPV test instead of Pap cytology for first-line primary screening in women 25 and older. Numerous HPV DNA tests are currently available (Table 2).

Another commonly used method in triage testing is the dual immunohistochemical (IHC) staining of p16INK4a and Ki-67. It is now believed that a positive staining for both is reliable evidence that a cell has been transformed due to a persistent HR HPV infection, increasing diagnostic certainty and enabling risk stratification [44,45,46]. On the other hand, it should be noted that the use of p16/Ki-67 IHC in younger women (<35 years) suffering from LSIL was significantly less effective than in older women [47].

Due to the viral origin of CC, prophylactic HPV L1 virus-like particle vaccines were introduced in 2009; namely, the bivalent Cervarix vaccine against HPV 16 and 18 and the quadrivalent Gardasil vaccine against HPV 6, 11, 16, and 18. Later, the nonavalent vaccine Gardasil 9 (HPV types 6, 11, 16, 18, 31, 33, 45, 52, and 58) was developed, which is now the only vaccine used in the U.S. These vaccines are most useful when applied to both girls and boys at an early age (11 or 12 years) but it is recommended for up to 26 years of age [48]. Older age groups have lower benefits due to an active sexual life and, thus, a higher probability of prior virus exposure. Clinical trials have demonstrated that HPV vaccines are highly effective in preventing HPV infection, but only before the first exposure to the virus [49]. Moreover, HPV vaccines have been found to reduce infections in anal [50] and oral regions [51]. The trials for the approval of the second-generation vaccine Gardasil 9 were found to be safe and almost 100% effective in preventing cervical, vulvar, and vaginal infections and precancers caused by all targeted HPV types [52]. Vaccinations still represent the only preventive tool in cervical cancer elimination, yet it faces challenges such as high costs, the requirement for multiple doses, a lack of access in developing countries, and a lack of community engagement.

The standard management of CC includes a sequence of cytology and HPV triage (although HPV test implementation differs across countries), followed by a colposcopy (a microscope-aided inspection of the surface of the cervix) [53], biopsy, and histological examination to confirm the diagnosis and help treatment decisions. When HSIL is confirmed, the WHO guidelines recommend various treatment options, but conization by a large loop excision of the transformation zone is currently preferred [54]. These WHO recommendations apply to women > 30 years of age but may extend to younger women with a high risk of HSIL. The WHO gives priority to the screening of women aged 30–49 years, rather than maximizing the number of screening tests in a woman’s lifetime. The early recognition of the disease is very important, enabling treatment in the precancerous stage when treatment is minimally burdensome with an excellent prognosis and can be done on an outpatient basis. The combination of early screening with vaccinations could drastically reduce the number of CC cases and deaths each year, making CC a preventable disease.

### 3.2. Novel Biomarkers in Liquid-Based Cytology

Despite the usefulness of the above-mentioned tests, novel biomarkers have been investigated that would reduce the colposcopic referral rate for those women with HR HPV infections that are unlikely to progress to invasive CC. Probably the most promising biomarker seems to be the detection of the HPV *E6* and/or *E7* mRNAs, elevated levels of which, in cervical samples, indicate increased viral activity and thus the potential presence of a transforming infection. For instance, a Korean study from 2014 demonstrated the sensitivity and specificity of the RT-qPCR assay for mRNA detection to be 91% and 98.6%, respectively, when analyzing HSILs [55].

A more recent class of biomarkers that is not currently applied in clinical practices include various types of short or long non-coding RNAs, altered DNA methylation patterns, or the gain of chromosome 3q. For example, mounting evidence suggests that miRNAs show specific expression profiles at various stages of cervical pathology. However, there is currently no established algorithm that would incorporate the alterations in miRNA expressions to CC screening, as discussed by Pisarska et al. [56]. In the same review, the authors comprehensibly described the relation of miRNA profiles with HPV infection. Another class of non-coding RNAs, lncRNAs, have emerged as potential biomarkers with interesting applications, including the detection of minimal residual diseases, auxiliary staging, real-time drug resistance monitoring, predicting the risk of metastatic relapse, and patient prognosis [57,58,59]. Most of the literature describes two circulating lncRNAs in blood with potential diagnostic and prognostic applications in CC (PVT1 and HOTAIR), but neither has shown significant upregulation patterns for them to be considered as reliable biomarkers [59].

DNA methylation is another epigenetic mechanism that has been extensively investigated in the process of cervical carcinogenesis. Most studies used LBC samples to examine the methylation profiles of promoters of tumor suppressor genes in CC [60,61], as well as LSILs and HSILs [62,63]. Methylation rates usually correlated with the disease stage (being highest in invasive carcinomas). In precancerous lesions, frequently observed differences in the rate of methylation between LSILs and HSILs were reported, e.g., for the promoters of the *SFRP* gene family [64], namely *CDKN2A* [65], *HS3ST2* [66], *CADM1*, *MAL* [67,68], *DAPK* [69], and *SOX1* [70]. Besides human genes, the methylation of HPV DNA has also been proposed as a novel biomarker for the triage of HPV-positive women. Higher HPV methylation was associated with increased disease severity, especially for the HPV *L1* gene (reviewed in [71]). Interestingly, we observed the opposite trend for the *E6* gene promoter where gradual demethylation correlated with the progression of precancerous lesions [72]. This is not surprising given that the E6 protein acts as an oncogene and not as a tumor suppressor. Despite numerous studies, a reliable and widely accepted methylation panel of genes that would improve the clinical performance is yet to be established.

The amplification of chromosome 3q has been consistently observed in both HSILs and invasive cervical squamous cell carcinomas. Chromosome arm 3q contains a human telomerase gene in region 3q26, and the overexpression of this gene, especially the catalytic subunit of human telomerase reverse transcriptase (*hTERT*), is one of the crucial steps for malignant transformation. Several studies have shown that the gain of chromosome 3q26 is the most consistent genetic abnormality in HSIL [72,73,74,75,76,77], and the frequency of the 3q26 gain has been shown to increase with the severity of the disease [78]. A study by Heitmann et al. demonstrated a sensitivity of 80%, a specificity of 90%, a negative predictive value (NPV) of 98%, and a positive predictive value (PPV) of 44% for the automated scanning for the 3q26 gain among women with LSIL cytology at colposcopy [73]. More prospective studies are underway to evaluate this biomarker for LSIL detection.

### 3.3. The Analysis of Circulating DNA in Cervical Precancerous Lesions

Studies that detect circulating diagnostic biomarkers of CC are less frequent, which is not surprising given that the cervix is easily accessible and thus a blood analysis seems unnecessary. In fact, most authors focus on the role of HPV-circulating DNA as a prognostic biomarker in the blood of patients with primary tumors to monitor the advanced stages or possible metastases [79,80,81,82,83], but not to analyze precancerous lesions. It was reported that non-metastatic CC patients with circulating HPV DNA in plasma had a tendency towards poor clinical outcomes and the development of recurrent distant metastases [80]. A recent meta-analysis confirmed that circulating HPV DNA in patients with CC could be used as a noninvasive dynamic biomarker of this type of tumor, with high specificity but only moderate sensitivity [84].

One of the few studies analyzing circulating HPV DNA in precancerous lesions is by Cocuzza et al. who evaluated whether circulating HPV DNA could be detected in the plasma samples of 120 women with regressed lesions, ASCUS, LSILs, and HSILs [85]. The authors found 53/120 samples to be positive for one of seven HR HPV types (HPV 16, 18, 31, 33, 45, 51, and 52), but only 41/120 samples were HPV-positive when analyzing the plasma of the same patients. The concomitant detection of the seven HR HPV types investigated in both cervical and plasma samples increased with the severity of the disease, ranging from 15.4% (4/26) in women with normal cytology/regressed lesions to 38.9% (7/18) in women with HSIL. Although the authors concluded that circulating HPV DNA in plasma samples could represent an interesting diagnostic biomarker for precancerous lesions, further studies are required to confirm their findings.

Another option to non-invasively monitor cancer markers is to analyze tumor-specific mutations in the ctDNA shed by a primary tumor into the bloodstream, which is an approach often used for other tumor types, but not in CC, where other options are available (as described above). Nevertheless, a study by Lee et al. reported a large screening of the mutations in CC from ctDNA using NGS on a panel of 24 genes [86]. Results showed that 18 of the 24 genes in the NGS panel had mutations across 24 CC patients, including somatic alterations of the mutated genes (*ZFHX3* in 83%, *KMT2C* in 79%, *KMT2D* in 79%, *NSD1* in 67%, *ATM* in 38%, and *RNF213* in 27%). Moreover, the authors concluded that the *RNF213* mutation could be useful for monitoring responses to chemotherapy and radiotherapy. Furthermore, in a study by Charo et al. NGS was applied for the analysis of ctDNA in the plasma of patients with gynecological malignancies of the cervix, ovary, or uterus [87]. The plasma of thirteen CC patients exhibited mutations in *PIK3CA* (*N* = 8, 61.5%), *TP53* (*N* = 5, 38.5%), *FBXW7* (*N* = 3, 23.1%), *ERBB2* (*N* = 2, 15.4%), and *PTEN* (*N* = 2, 15.4%). However, these studies did not include women with precancerous lesions (LSILs or HSILs), and it is therefore difficult to say which gene mutations are potentially useful predictors of precancer progression.

## 4. Endometrial Precancerous Lesions and Early-Stage Endometrial Cancer

### 4.1. Endometrial Cancer

Endometrial cancer (EC) is the most common cancer of the female genital tract in developed countries and is the sixth most common cancer in women worldwide, with more than 400,000 new cases diagnosed in 2020 [88]. Both its incidence and its associated mortality are increasing [89]. In routine clinical practices, EC is classified into type I or type II, based, in particular, on the histology and grade. Type I (endometrioid) is more common (80–85% of cases) and includes low-grade, diploid, hormone receptor–positive endometrial tumors with a good prognosis. Common molecular alterations identified in type I tumors are microsatellite instability, as well as *PTEN*, *KRAS,* and *CTNNB1* mutations [90]. Type II (non-endometrioid, serous) represents high-grade, usually aneuploid, hormone receptor–negative endometrial tumors frequently associated with a poor prognosis and an increased risk of metastasis development. In this type we often encounter *TP53* mutations, Her-2/*neu* amplifications, negative or reduced E-cadherin expressions, and the inactivation of p16 by mutation or hypermethylation. However, the most important limitation of the classification of endometrioid and serous carcinomas is that the categorizations and behaviors often differ from the theory in specific cases [91]. Recent molecular studies found that EC comprises a range of diseases with distinct genetic and molecular features. Four novel EC categories have recently been proposed by The Cancer Genome Atlas Research Network (TCGA), such as polymerase epsilon (*POLE*), ultra-mutated, microsatellite unstable (MSI), copy-number low, and copy-number high [92]. This new genomic-based characterization evoked the reclassification of EC, which has recently led to the updating of the European guidelines for diagnosis and treatment [93]. However, significant clinical issues, such as the more detailed stratification of the non-specific molecular profile of EC, still remain to be solved. Thus, further studies should be focused on the integration of molecular and clinicopathological features [94].

### 4.2. EC Development: Endometrial Precancer–Cancer Sequence

The human endometrium is a highly regenerative tissue, adopting multiple different physiological states during life. During the reproductive years, the endometrium undergoes monthly cycles of growth and regression in response to oscillating levels of estrogen and progesterone sustained by stem/progenitor cells [95]. Thus, an increased rate of mutations in this tissue can be expected, and it mirrors the fact that EC is one the most common gynecological tumors. Accordingly, normal endometrial glands (over 50%) frequently carry ‘driver’ mutations in cancer genes such as *KRAS*, *PIK3CA*, *FGFR2,* and/or *PTEN* loss, the burden of which increases with age [96]. Whole-genome sequencing showed that normal human endometrial glands are clonal cell populations with total mutation burdens that increase at about 29 base substitutions per year; however, these are many-fold lower than those of EC [97]. Interestingly, the extremely high mutation loads attributed to the DNA mismatch repair deficiency and *POLE* mutations, as well as structural and copy number alterations, are specific to EC, not to normal epithelial cells [98]. Comprehensive examination of the timing of pathogenic somatic *POLE* mutations in sporadic endometrial tumors by whole genome sequencing confirmed that pathogenic somatic *POLE* mutations occurred early, and are possibly initiating events in endometrial and colorectal tumorigenesis [99]. It was also shown that the acquisition of a *POLE* mutation caused a distinct pattern of mutations in cancer driver genes, a substantially increased mutation burden, and an enhanced immune response, detectable even in precancerous lesions.

In a recently published study, Aguilar et al. identified the presence of driver mutations (e.g., in *KRAS, PTEN, CTNNB1*, etc.) by high-throughput sequencing of serial endometrial biopsies taken several years before the onset of EC, even without previously diagnosing atypical hyperplasia or endometrioid intraepithelial neoplasia (EIN). It is important to note that these mutations were confirmed in the invasive cancer. This research provided unique insights into precancer initiation and progression and clearly demonstrated the existence of endometrial premalignant lesions with definitive mutations not readily identifiable by histology [100]. These findings are also supported by a case report in which tumor-specific mutations were identified in an asymptomatic individual without clinical or pathologic evidence of cancer nearly one year before symptoms developed, i.e., postmenopausal bleeding and a single microscopic focus of EC diagnosed at the time of hysteroscopy [101].

From the histological point of view, endometrial hyperplasia represents a spectrum of irregular morphological alterations, whereby the abnormal proliferation of the endometrial glands results in an increase in the gland-to-stroma ratio compared to the endometrium from the proliferative phase of the cycle. Nevertheless, different types of EC derive from different precursor lesions. Type I EC typically develops from atypical hyperplasia or EIN, depending on the classification system [102]. It was shown that the risk of progression to carcinoma in women with non-atypical endometrial hyperplasia was <5%, while almost 30% of women with atypical endometrial hyperplasia were diagnosed with EC [103]. Moreover, up to 50% of women with atypical hyperplasia on an endometrial biopsy have EC in the resection specimen [104]. Type II EC usually develops on the atrophic endometrium, often on an endometrial polyp. This lesion is composed of cytologically malignant cells, like those seen in uterine serous carcinoma, lining the surface of the endometrium or endometrial glands without the invasion of the endometrial stroma, myometrium, or lymphovascular spaces [105]. Although technically non-invasive in appearance, these tumors have been associated with extrauterine disease, reflecting their aggressive biology [106]. However, there are many unanswered questions, and it is not clear if cancers obey these model paradigms in all cases.

Taken together, it can be assumed that endometrial precancerous lesions preceding the development of invasive ECs are molecularly different from the normal endometrium, frequently show a monoclonal growth pattern, and share some, but not necessarily all, features of a malignant endometrium [107]. Indeed, these molecular alterations, predominantly single hotspot cancer driver mutations in the context of suspicious histopathologic features, should be detectable already in precursor lesions, especially those with an increased risk of progression to EC and could have sufficient sensitivity and specificity.

### 4.3. The Current State of the Diagnosis and Screening of Endometrial Precancer and/or Early EC

Although the detection of precancerous lesions, and the patient risk stratification in general, are increasingly important for early diagnoses and the prevention of cancer, the screening of EC and precancerous dysplasia in the non-symptomatic population basically does not exist (Table 2). In most cases, EC is diagnosed in symptomatic women. There are several lines of evidence that a diagnosis of endometrial hyperplasia may precede the development of endometrioid EC, as they share common environmental predisposing risk factors, such as a postmenopausal status, family history, nulliparity, obesity, diabetes mellitus, long-term Tamoxifen therapy, elevated estrogen levels, smoking, etc. [108,109]. Known genetic risk factors for endometrial cancer are germline mutations in DNA, mismatch repair genes associated with Lynch syndrome [110], and germline *PTEN* mutations responsible for Cowden syndrome [111]. Hereditary EC makes up approximately 2% to 5% of all cases [112]. For women who tested negative for hereditary mutations, the concept of a polygenic risk score (PRS) based on genetic variants determined by genome-wide association studies would be of potential interest, since it predicts women who are at high risk of developing EC [113]. Moreover, the integration of an EC PRS with other known EC risk factors (e.g. obesity) should improve the risk stratification accuracy and could provide opportunities for population-based screening [114].

In current practices, annually performed clinical examinations and transvaginal ultrasounds are insufficient. On the other hand, an additional endometrial biopsy and outpatient hysteroscopy could improve screening results but are not well tolerated. Pipelle sampling can be limited in cases with a non-representative biopsy specimen, or cervical stenosis [4]. This has made EC and its precancerous lesions a major issue for health care investigations. Already, G. Papanicolaou has been interested in the possibility of the diagnostic value of cervical smears. Due to the anatomical continuity of the uterine cavity within the cervix, material from routine cervical Pap brush samples represent a unique opportunity to detect the signs of disease shed from the upper genital tract [2]. However, a systematic review by Frias-Gomez et al. showed that only approximately 40–50% of ECs can be detected with a morphological evaluation of cervical smears [115]. A recent retrospective study, focused on the sensitivity of cervical cytology in EC detection, showed a sensitivity of only 25.6% [116]. Cervical Pap brush samples were also subjected to PapSEEK, the targeted sequencing of specific PCR amplicons generated from the specific loci of 18 genes, which resulted in the detection of 81% EC-positive patients and 78% early-stage EC cases [8].

A recently published pilot cytological analysis of self-collected voided urine samples and vaginal samples collected with a Delphi screener before routine clinical procedures reported that the combination of non-invasive urogenital sampling with cytology distinguished malignant from non-malignant causes of postmenopausal bleeding, with approximately 90% accuracy [14]. Following these promising results, the multicenter prospective validation study (DETECT study, ISRCTN58863784) has been launched to support the utilization of this non-invasive test in clinical practice [117].

Proteins, rather than genes or RNAs, perhaps reflect the properties of living tissue most accurately. Thus, proteomics has emerged as the technique of choice for biomarker discovery. A number of blood-based biomarker candidates for EC detection have been reported, belonging to hormones, cancer-associated antigens, adipokines, complement factors, plasma glycoproteins, plasma lipoproteins, enzymes and their inhibitors, growth factors, etc. (for detailed information, see the review by Njoku et al. [118]). Urine represents an attractive biofluid for biomarker discovery and, indeed, elevated levels of the zinc-alpha-2 glycoprotein, alpha-1 acid glycoprotein, and CD59 indicating the presence of EC were identified by proteomics [119]. Several IHC biomarkers have been also investigated in combination with hematoxylin and eosin (H&E) staining to improve the diagnostics of endometrial precancers and to predict the risk of the transition from hyperplasia to EC. There are several prominent IHC candidates, such as PTEN, p53, PAX2, beta-catenin, E-cadherin, and proteins involved in the DNA mismatch repair pathways (MLH1, MLH2, and MSH6) that may be useful in predicting malignant progression [120].

Nowadays, genomic analyses offer an excellent opportunity to stratify the risk of EC progression; however, despite the considerable worldwide incidence of EC, there is currently no blood-based biomarker in routine use for EC patients [121]. Circulating tumor DNA has been recently shown as an important source of genetic information that may enable the detection of both early- and late-stage EC. An NGS analysis of ctDNA from peripheral blood plasma revealed that a mutation in at least one of the four genes, *PTEN, PIK3CA, KRAS*, and *CTTNB1*, can be detected in more than 90% of EC patients [121]. Notably, several studies have determined ctDNA in different body fluids and have confirmed its clinical utility for EC patients. For instance, an NGS analysis of uterine aspirates, in combination with an analysis of ctDNA and CTC obtained from blood samples, clearly indicate the potential clinical applicability [122].

Efforts focused on the investigation of circulating free miRNAs as potential biomarkers of early EC are also growing, since these miRNAs have been described as potential biomarkers for these malignancies [123]. Interestingly, several studies have identified miRNAs in extracellular vesicles from different body fluids, e.g., urine [124], peritoneal lavage [125], and blood serum [126]. Recently, a large plasma-derived exosomal miRNA study identified miR-15a-5p as a valuable diagnostic biomarker for the early detection of EC [127]. Another study identified four miRNAs associated with EC (oncogenic miRNAs miR-135b and miR-205, as well as tumor suppressor miRNAs miR-30a-3p and miR-21) [123].

Metabolites represent another promising source for the detection of early-stage endometrial cancer and can be detected in endometrial tissue, brush and lavage specimens, blood samples, and urine [2]. Blood metabolites are of great interest since they are easily accessible, although they have a limited yield. Several blood-based metabolites have been suggested as potential EC biomarkers and are mostly by-products of lipids and amino acids [128]. Interestingly, the most commonly reported dysregulated metabolic pathways responsible for the presence of biomarkers in the serum of EC patients are the lipid- and glycolysis-related pathways [129,130]. One of the most promising is phosphocholine, whose elevated levels (approximately a 70% increase) have been identified in EC patients [131]. Recently, Njoku et al. showed that the determination of specific lipid metabolites in blood, such as phospholipids and sphingolipids, could enable the early detection of EC [132].

## 5. Strategies for the Early Detection of Epithelial Ovarian Cancer (EOC)

### 5.1. Current EOC Screening

In 2020, EOC was the eighth most common cancer in women worldwide, with more than 300,000 new cases and more than 200,000 deaths per year (https://gco.iarc.fr/, accessed on 22 June 2021). It is therefore of critical importance to diagnose ovarian tumors at early stages. Stage I tumors can be cured in up to 90% of patients using the currently available surgery and chemotherapy, and five-year survival in stage II tumors can exceed up to 70%. However, once the cancer reaches stages III or IV, treatment success drops to 25–30% or less [133]. In the absence of an effective screening strategy, only 25–30% of ovarian cancer is diagnosed at an early stage (FIGO I-II). Therefore, a sufficiently sensitive and specific screening strategy is urgently needed [134].

Most screening strategies for the detection of ovarian tumors are based on the detection of the serum biomarker CA-125 and transvaginal sonography (TVS). The separate use of these methods has a very low sensitivity and specificity, as evidenced by several studies [135,136]. Therefore, a two-step strategy, using both CA-125 and TVS, has been developed. Several studies have used a two-phase strategy to detect early-stage ovarian tumors and have confirmed its good specificity and positive predictive value in a population of women at a moderate risk of ovarian cancer [3,127,128].

Genetic factors play an important role in the susceptibility to EOC. The risk of the disease is known to be three times higher in first-degree relatives of patients with EOC than in the general population [137]. Approximately 25% of the familial relative risk (FRR) is represented by high-penetration mutations in *BRCA1* and *BRCA2* [138]. Another 10% is explained by moderate risk mutations in *MLH1*, *MSH2*, *MSH6*, *RAD51C*, *RAD51D,* and *BRIP1* [139], and 6.4% of the FRR is attributed to about 30 common low-risk single nucleotide polymorphisms (SNPs) identified by genome-wide association studies in relation to EOC [140,141]. Although each individual SNP is associated with only a low risk of EOC, in combination, their effects on risk may be greater. The evaluation of the combined effect of genetic variants using a polygenic risk score (PRS) is therefore beginning to gain ground as a means of distinguishing patients at a high and low risk of developing the disease [142]. Currently, germline mutations of *BRCA1* and *BRCA2* are being monitored, which are associated with a higher lifetime risk of ovarian cancer compared to the general population [137]. Because there is no reliable strategy for the early detection of tumors in this group of women, a bilateral salpingo-oophorectomy with a hysterectomy is recommended to reduce the risk of EOC development after a woman completes her reproductive plans. Another option is the monitoring of CA-125 levels, in combination with a TVS examination, every six months. According to the presence of malignant lesions, it is estimated that up to 70% of “ovarian” carcinomas that develop in women carrying *BRCA1* or *BRCA2* mutations, but also in women at a conventional genetic risk, may arise not from the ovaries but rather from oviduct fimbriae, which represents an even greater challenge for early detection [143].

### 5.2. Protein Biomarkers in Liquid Biopsies

The use of liquid biopsies seems to be ideal for the detection of the early stages of EOC. The level of the mentioned CA-125 is already commonly determined from blood at this time. Nevertheless, significant levels of CA-125 have been detected in only 80% of EOC. Thus, if CA-125 alone is used during the initial screening, at least 20% of EOC is not found [144,145]. The aim of the current studies is to find biomarkers that, in combination with CA-125, would increase its sensitivity in detecting the early stages of the disease without reducing specificity. Published studies have described many different biomarkers that improved the sensitivity of CA-125 in the early phase of disease detection [146,147], but the usefulness of these combinations has not yet been confirmed by independent validation.

Much attention has been paid to the combination of CA-125 and the human epididymis 4 (HE4) protein. HE4, also known as the whey acidic protein (WFDC2), is slightly less sensitive to the detection of early-stage ovarian cancer compared to CA-125 but it has better selectivity for distinguishing between malignant and benign pelvic masses. CA-125 and HE4 have been applied in combination to classify patients for specialized surgery and are used in the malignancy risk algorithm ROMA (Risk of Ovarian Malignancy Algorithm) [148] or are supplemented by three other biomarkers (transferrin, apolipoprotein A1, and follicle stimulating hormone) in the OVERA test [149], which is also used to distinguish between malignant and benign pelvic masses. ROMA combines the serum levels of CA-125 and HE4 together with the menopausal status in a logistic regression model and classifies pelvic mass patients into groups with a high or low risk of EOC [150]. In test studies with a specificity of 75%, ROMA showed a sensitivity of 94% in distinguishing benign tumors from EOC with an 85% sensitivity to the identification of the early stages (I and II) of disease [148].

Another possibility to detect early stages of EOC using liquid biopsies is via autoantibodies that have developed against proteins associated with the cancer. Mutations in the *TP53* gene have been identified in almost all high-grade serous tumors, and autoantibodies against p53 have been reported in approximately 20–25% of cases. Yang et al. studied autoantibodies against p53 in the serum of women who donated blood months to years before their diagnosis, and found that elevated anti-p53 antibodies could be detected on an average of 8 months before CA-125 elevation, and 22 months before the clinical diagnosis in patients with no increase in CA-125 [151]. Several other autoantibodies have been identified that may be useful for screening, including the anti-prostaglandin F receptor (anti-PTGFR) and the anti-protein tyrosine phosphatase type A receptor (anti-PTPRA). These antibodies are detected in 22–32% of ovarian cancers at a 95% specificity [152]. Another study reported a panel of autoantibodies to p53, namely, TRIM-21, NY-ESO-1 (CTAG-1A), and PAX-8, which achieved a sensitivity of 46% to 56% with a specificity of 98% [153].

Based on previous studies, Guo et al. found that the combination of CA-125, osteopontin, the macrophage inhibitory factor, and the anti-IL8 autoantibodies were able to detect 82% of early-stage EOC (compared to 64% for CA-125 alone) with a 98% specificity [154]. Similarly, the combination of CA-125 and HE4 autoantibody complexes increase the proportion of detected cases to 81% [155]. Ma et al. developed a non-invasive serological diagnostic approach using protein microarrays, identifying and evaluating a set of candidate autoantibodies to EOC-associated antigens. They found that the optimized panel of three autoantibodies (GNAS, p53, and NPM1) had relatively high sensitivity (51.2%), specificity (86.0%) and accuracy (68.6%). This panel was able to identify 51% of CA-125-negative patients with EOC [156].

### 5.3. Circulating Tumor Cells

Research on CTC in EOC initially received little attention since the direct peritoneal spread in the abdomen is considered the primary route of metastasis in EOC, and distant metastases occur in only one-third of patients. Therefore, it was assumed that there was an insufficient entry of tumor cells into the bloodstream. The amount of CTCs are mainly related to tumor progression, CA-125 levels, or residual disease after surgery and may, thus, have some prognostic or predictive value. However, the use of CTCs in EOC is affected by their isolation and detection methods [157].

CTCs are commonly enriched in two ways, cell surface marker-dependent and marker-independent approaches. As we discussed before, the most often used approach is based on capturing epithelial cell adhesion molecule (EpCAM), a marker on the surface of epithelial cells which is absent in normal leukocytes [25]. This marker is used in the CellSearch System, which separates epithelial cells from blood using EpCAM-associated ferrofluids, and then identifies them using fluorochrome-bound antibodies (cytokeratin-positive and CD45-negative) [158]. Using the CellSearch System, a detection rate for EOC of between 12% and 90% was achieved. However, the rate of the detection of CTCs in the early stages of the disease was generally low and the findings were controversial. Moreover, the number of CTCs in patients with EOC as determined by CellSearch did not correlate with clinical characteristics or patient outcomes [159]. In contrast, marker-independent CTC isolation approaches that utilize the biophysical properties of CTCs, such as their size, deformability, and dielectric susceptibility achieved more promising results in EOC detection [160]. Guo et al. examined the diagnostic value of CTCs in peripheral blood in 61 patients with suspected ovarian tumors. CTCs were identified and calculated by microfluidic isolation and the immunofluorescence staining of CD45, HE4, and epithelial and mesenchymal markers (EpCAM, cytokeratins, and vimentin). This method demonstrated 73.3% sensitivity and 86.7% specificity, and could serve as a useful diagnostic indicator in patients with suspected ovarian cancer [161].

### 5.4. Circulating Free DNA and Circulating Tumor DNA

The potential use of cfDNA and ctDNA in EOC has been widely described in the last twenty years, including the use of screening, predicting responses to systemic therapies, and monitoring subclinical diseases. These works were analyzed in several review papers, mapping the methods, procedures, and results obtained so far [162,163].

A meta-analysis by Thusgaard et al. included 36 relevant studies that evaluated the use of ctDNA for tumor detection, the early diagnosis of disease, and monitoring the response to treatment. However, the individual studies differed significantly in the methods used to analyze ctDNA, and in their sample size. The overall conclusion was that ctDNA may be a promising biomarker for EOC diagnosis, prediction, and prognosis. However, more studies are needed to identify proper methods and settings for the clinical use of ctDNA [164]. Interestingly, only one study on the detection of the early stages of the disease that was included in the meta-analysis that analyzed the methylation of cell-free DNA [165]. Other uses of ctDNA methylation for early diagnoses are summarized in a review by Guo et al. which analyzed the results of 18 relevant studies. The main problem was, again, the considerable heterogeneity and the different numbers of samples studied. These studies included a wide range of genetic targets, including classical tumor suppressor genes (*BRCA1* and *PTEN*) and EOC-specific tumor suppressors (*RASSF1A* and *OPCML*). Nevertheless, because most genes were included in only one or two studies, the ability to perform a gene-level analysis was very limited. In addition, none of the genes were identified as predominantly methylated for ovarian tumor tissue. However, these data suggest that serum/plasma cfDNA methylation assays can achieve a robust diagnostic accuracy in EOC (with a median of 85% and a range of 40–91%), especially when multiple genes are used, and ovarian tumors are compared to benign pelvic masses. However, even here it is necessary to optimize the range of possible gene targets and techniques, and to include more early-stage EOC samples [166].

### 5.5. Exosomes and Circulating Cell-Free MicroRNAs

To date, several exosomal markers for EOC have been described, such as CD24 and EpCAM [167]. Zhao et al. reported that the combination of three exosomal markers, CA-125, EpCAM, and CD24, which can provide the required diagnostic accuracy for the early diagnosis of EOC [168]. A strong correlation between tumor cell mRNA, miRNA profiles, and EX was confirmed. Since miRNA expression is dysregulated in most cancers, different types of cancer represent different exosomal signatures of miRNAs [169].

Taylor and Taylor distinguished benign cases of EOC from patients with different stages of EOC by profiling eight microRNAs in EpCAM-positive EX that was isolated from peripheral blood [170]. Yoshimura et al. reported that the level of exosomal miR-99a-5p is significantly up-regulated in the serum of patients with EOC [171]. Similarly, other proteins, e.g., TGF-β1 [172], membrane protein Claudin-4 [173], L1CAM, CD24, ADAM10, and EMMPRIN [174] have been described as exosomal biomarkers for the early diagnosis of EOC. In addition to peripheral blood, urine can also be used as a source of circulating EX. Zhou et al. found that the urinary level of exosomal miR-30a-5p in patients with ovarian serous adenocarcinoma was 3.3-fold higher than in healthy women [175]. A recently established integrated exosome profiling platform (ExoProfile chip), which is based on the diagnostic power of seven biomarkers (HER2, EGFR, FRα, CA-125, EpCAM, CD24, and CD9) plus CD63, showed promising results for distinguishing benign tumors and recognizing the early and late phases of EOC [176].

## 6. Conclusions

The mutational landscape of cancer is generated through a combination of environmental and endogenous stresses that cause base substitutions, insertions, deletions, and chromosomal rearrangements. In many tumors, this is a consequence of specific defects in the cellular processes responsible for maintaining genomic integrity [177]. These alterations are not only the cause of the disease but can also serve to detect cancer in its early stages. Liquid biopsy strategies are promising, as they are minimally invasive approaches that are mainly based on the characterization of circulating tumor DNA, circulating tumor cells, and circulating extracellular vesicles as sources of proteomic and genetic information mirroring alterations in transformed cells.

The current practice in prevention, screening, diagnostics, and treatment of all three gynecological malignancies are summarized in Table 3. The situation in CC meets the main task of the Lancet Oncology Commission, which is to focus attention on prevention [178]. This comprises vaccinations [49] and well-organized screening programs, including cytology with an HPV testing triage that enables early diagnostics by expert colposcopies and biopsies, followed in most cases by early treatment in the precancerous stages of the disease. This combination has been responsible for a dramatic decline of invasive CC incidence and mortality. If all these conditions are met, cervical cancer could become a preventable disease [53]. In contrast, endometrial and ovarian cancers are diagnosed only when symptoms first appear. In these tumors, diagnostics are limited by the relatively poor accessibility of the uterus and ovaries for sampling. In EC, for instance, expectations placed on the transvaginal ultrasound as a screening method were not fulfilled [179]. An emphasis is thus given to women at a higher risk, i.e., characterized by obesity, diabetes, hypertension, or Lynch syndrome. The disease is diagnosed when the first symptoms occur, usually bleeding. When curettage is performed immediately and the disease is confirmed, operative treatments show good results. On the other hand, neither an effective screening method, nor the ability to accurately define a high risk group are available for OEC, which means that > 75% of women are diagnosed with an advanced stage disease [180]. Whilst surgical improvements followed by systemic and biologic targeted therapies will be useful for existing patients, the discovery and implementation of new diagnostic and/or prognostic biomarkers, particularly those detectable in liquid biopsies, will be important for early detection and improved patient outcomes.

## Figures and Tables

**Figure 1 cancers-13-06339-f001:**
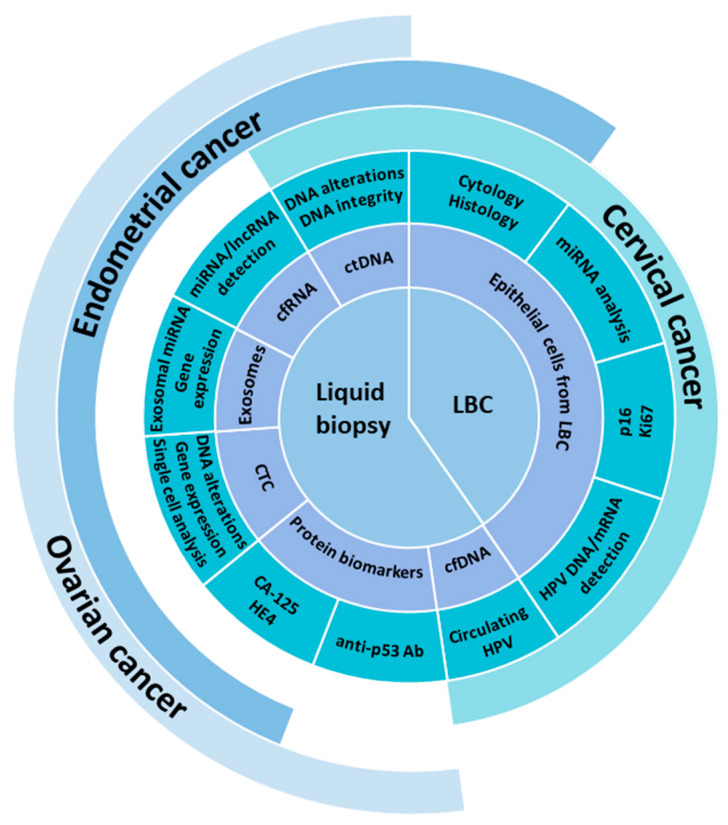
Application of liquid biopsies and liquid-based cytology (LBC) for diagnostics in clinical practice and/or research.

**Figure 2 cancers-13-06339-f002:**
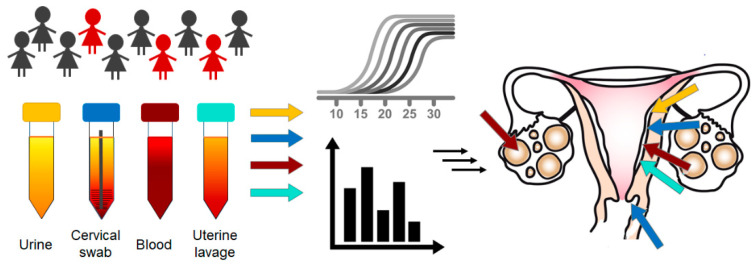
Various sources of body fluids are used for detection of circulating biomarkers in non-invasive format to identify women at high risk of cervical, endometrial and ovarian (pre)cancer.

**Table 1 cancers-13-06339-t001:** Overview of selected approaches using minimally invasive gynecological sampling methods for the detection of ovarian, endometrial, and cervical malignancies.

Source	Biomarker	(Pre)Cancer	Application and References
Pap smear/Pap test/PapSEEK test	DNA	CC EC OEC	Pap test as a screening method for early-stage cervical, ovarian, and endometrial cancers [7,8]
Uterine lavage	DNA	EC OEC	Uterine lavage fluid is used to detect early endometrial carcinomas by genomic analysis [9]; DNA sequencing from uterine lavage samples with a focus on a panel of candidate ovarian cancer driver genes, including *TP53* [10] and *MCM5* [11].
Urine	Protein biomarkers Cytology miRNA HPV DNA	OEC EC CC	MicroRNA (miR-223, let7-i, miR-34a, and miR-200c) expression levels in urine as a non-invasive diagnostic test for endometrial cancer [12]; summary of studies testing the level of HE4 in urine and body fluids for the diagnosis of gynecological cancer [13]; cytology of urine and vagina fluids as a sensitive and specific tool for the detection of gynecological tumors [14]
Cervicovaginal secretions	Metabolomic biomarkers Protein biomarkers Cytology	OEC EC CC	Phosphocholine, asparagine, and malate from cervicovaginal fluid have been identified by nuclear magnetic resonance spectroscopy as promising metabolomics biomarkers for EC detection [15]; alpha-actinin-4 as a promising biomarker for the detection of precancerous state of cervical cancer [16]; CA-125 in cervicovaginal secretion as a potential biomarker for EC detection [17]; cytology of urine and vagina fluids as a sensitive and specific tool for the detection of gynecological tumors [14]
Endocervical swabs	Protein biomarkers Enzymatic activity HPV DNA	EC CC	Proprotein convertase activity [11,18]; HPV DNA testing from cervical swabs as an alternative mechanism to routine cytological screening [19]
Tampons or vaginal swabs	Protein Biomarkers HPV DNA	CC	Comparison of vaginal swabs and urine samples with cervical smears for HPV testing in cervical cancer screening strategies [20,21]

**Table 2 cancers-13-06339-t002:** Commercially available tests and assays for diagnostics of cervical, endometrial, and ovarian malignancies.

Test	Vendor	Application
Cervical cancer		
ThinPrep^®^ Pap Test	Hologic (Marlborough, MA, USA)	Cervical smear taken into a liquid medium followed by computer evaluation of the specimen
SurePath Pap Test	Becton Dickinson (Franklin Lakes, NJ, USA)	A liquid-based Pap test used in the screening and detection of cervical cancer, pre-cancerous lesions, atypical cells, and all other cytological categories
Roche Cobas^®^ HPV	Roche (Basel, Switzerland)	A qualitative in vitro test for the detection of HPV in patient specimens by amplification of target DNA and its hybridization for the detection of 14 high-risk HPV types
Cervista HPV16/18 assay	Hologic (Marlborough, MA, USA)	A qualitative, in vitro diagnostic test for the detection of DNA from two high-risk HPV types: 16 and 18
Hybrid Capture 2	Qiagen (Hilden, Germany)	The platform for the nucleic acid hybridization assay for the detection of HPV, *Chlamydia trachomatis,* and *Neisseria gonorrhoeae*
Linear Array HPV	Roche (Basel, Switzerland)	Test for genotyping HPV in cervical biopsies and other formalin-fixed, paraffin-embedded specimens
INNO-LiPA^®^ HPV Genotyping Extra II	Fujirebio (Tokyo, Japan)	Line probe assay, based on the reverse hybridization principle, designed for the identification of 32 different genotypes of HPV
Endometrial cancer		
None available		
Ovarian cancer		
OVA1^®^ and OVERA^®^ tests	Aspira Women’s Health Inc(Austin, TX, USA)	In vitro diagnostic multivariate index assay that analyzes the serum levels of proteomic biomarkers
Elecsys HE4 assay	Roche (Basel, Switzerland)	Sandwich electrochemiluminescent immunoassay, which measures the amount of HE4 in a patient sample against a calibration curve
Elecsys^®^ CA 125 II	Roche (Basel, Switzerland)	Biomarker test to determine the amount of CA 125 protein in a blood sample
Bard1 Life Sciences test	BARD1 Life Sciences (Notting Hill, Australia)	Autoantibody test for early detection of ovarian, breast, and lung cancers

**Table 3 cancers-13-06339-t003:** Current practice in prevention, screening, diagnostics, and treatment of all three gynecological malignancies.

(Pre)Cancer	Prevention ^1^	Screening	Diagnostics	Treatment	References
Cervical	Vaccination	Pap test HPV triage	Colposcopy + histology	Conization	[53]
Endometrial	None ^2^	None	Transvaginal sonography Endometrial biopsy Hysteroscopy	Hysterectomy	[118]
Ovarian	None ^3^	None	Histology in advanced stages (>75% of cases) Transvaginal sonography ROMA (CA125 + HE4)	Radical surgery Systemic therapy Targeted therapy	[134]

^1^ In contrast to cervical cancer, effective prevention and screening in endometrial and ovarian cancers do not exist. There are, however, several prophylactic measures, albeit less specific, that may substitute existing causal preventive procedures. ^2^ Prophylactic measures in endometrial cancer include healthy lifestyle (physical activity, no smoking, weight loss, fruits, vegetables, and vitamins, reduction of consumption of fat and protein), medical surveillance (obesity, diabetes, metabolic syndrome, hyperestrogenism, genetic factors, hormonal preventive therapy) and, if genetically justified, even prophylactic hysterectomy [181,182]. ^3^ In ovarian cancer, improvement of healthy lifestyle by similar means as in EC can be helpful. Medical surveillance is represented by questionable ovulation blockade using birth control pills or by salpingectomy for anything other than oncologic reasons; the most common preventive procedure remains the prophylactic adnexectomy from a genetic indication after reproductive fulfillment [183,184].

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
