# Peer review of "New Trends in the Detection of Gynecological Precancerous Lesions and Early-Stage Cancers"

_cancers, 2021, doi:10.3390/cancers13246339_

Round 1

Reviewer 1 Report

I find the manuscript very interesting, as the team of authors has dealt with a topic that is very important for a gynecologist and oncologist. However, I think that the authors should consider whether or not to focus only on ovarian cancer as a cancer with difficult early diagnosis. I cannot agree and ask for a change in the manuscript, because the nonspecific symptoms of ovarian cancer cannot be compared to endometrial cancer. In cancer, the endoemtrial bleeding occurs in 90% and patients visit a gynecologist sooner or later. I do not understand a bit why the authors discuss here in the early diagnosis of cervical cancer, in this case prophylaxis and management of HPV positive should be discussed extensively. I expect figures and diagrams that will make the manuscript read better and clearer. I am asking for age-specific procedures for the management of pre-neoplastic and early stages of cervical cancer.

Author Response

Dear Editor,

We would like to thank both reviewers for their insightful and helpful comments and suggestions to improve our manuscript. Below, please find our detailed responses to each of them, highlighted in blue color.

R1:

I find the manuscript very interesting, as the team of authors has dealt with a topic that is very important for a gynecologist and oncologist. However, I think that the authors should consider whether or not to focus only on ovarian cancer as cancer with difficult early diagnosis. I cannot agree and ask for a change in the manuscript, because the nonspecific symptoms of ovarian cancer cannot be compared to endometrial cancer. In cancer, endometrial bleeding occurs in 90% and patients visit a gynecologist sooner or later.

We appreciate Reviewer’s opinion, but the main goal of this review was not to compare symptoms among ovarian, endometrial, and cervical cancer but to provide the reader with a comprehensive description of the current state in diagnostics and early detection of these most common gynecological malignancies. We believe that it is important to highlight that while cervical screening for precancerous lesions is well-established and relatively successful, less is so for the other two malignancies. These differences are emphasized throughout the manuscript.

Considering endometrial cancer, there is currently insufficient evidence to support screening in either the general population or high-risk women for the detection and treatment of premalignant disease. Prevention strategies are therefore urgently needed, particularly with respect to the emerging epidemic of EC associated with the expansion of obesity worldwide.

Early diagnosis has great potential to improve outcomes as treatment can be curative, especially for early-stage disease. While invasive testing is necessary for a tissue diagnosis, most women with postmenopausal bleeding have a benign explanation for their bleeding. Currently, thousands of women with postmenopausal bleeding undergo hysteroscopy and/or endometrial biopsy, invasive tests that are unpleasant, sometimes technically challenging, and painful or extremely painful for 30%–40% of women. Thus, the possibility of non-invasive, accurate, and cost-effective tools enabling identification of women with EC at the earliest possible stages that in parallel will exclude many women who do not have EC is of great interest.

I do not understand a bit why the authors discuss here the early diagnosis of cervical cancer, in this case, prophylaxis and management of HPV positive should be discussed extensively.

We would like to keep discussion of cervical cancer screening for the reasons given above. Nevertheless, we now added two extra paragraphs on prophylaxis and management of HPV-positive women into Section 3.1, as suggested.

I expect figures and diagrams that will make the manuscript read better and clearer.

One new figure and 2 tables were added.

I am asking for age-specific procedures for the management of pre-neoplastic and early stages of cervical cancer.

This is now given at the end of Section 3.1.

Reviewer 2 Report

Holcakova and colleagues, in the review entitled “new trends in detection of gynaecological precancerous lesions and early-stage cancer” aimed to summarise the role of liquid biopsy in the early detection of cervical, endometrial and ovarian cancers.

The review is an important one, providing some insights into the role of circulating biomarkers in the detection of gynaecological malignancies. There are, however, significant number of issues that need addressing before it can be suitable for publication in Cancers.

  1. The manuscript does little in providing readers with the evidence base on the current trends in the early detection of gynaecological cancers, in a readily available format. Can they summarise the studies providing the key evidence on the current trends for early detection in the various cancer sites in a table and grade the level of evidence where possible, discussing the pros and cons of each test.
  2. A number of key studies have been missed out and should be included in the review. See subsequent comments on those studies.
  3. The simple summary and main abstract both require significant consolidation and rephrasing.
  4. Introduction line 51: In addition to uterine lavages and peripheral blood, we know there have been endometrial cancer biomarker studies conducted in cervico-vaginal fluids (Cheng et al, metabolomics 2019, O Flynn et al, Nature Comms 2021), urine, and peritoneal lavage (Roman-canal et al 2019) that need highlighting.
  5. Introduction line 54: “degree and intensity of cancer” are not commonly used terms in describing cancers. Do you mean disease stage and grade? Please clarify or rephrase as appropriate.
  6. Introduction lines 51-57 have no references. Please reference as appropriate, especially when describing the potential roles of liquid biopsies.
  7. Section 2: Overview of circulating biomarkers: For a review focused on gynaecological malignancies, the anatomical continuity between the upper genital tract and the lower genital tract, allowing for the exploitation of ovarian/uterine derived biomarkers to be sampled based on non-invasive or minimally invasive sampling methodologies should be highlighted. See the review by Costas et al. See recent study by O Flynn et al.
  8. Section on endometrial cancer: Line 249: There is a more recent reference with global incidence of endometrial cancer over 400,000. See Sung et al 2021.
  9. Line 368: wrong reference. What is ref 91 doing here?: “Vitamin A abuse in cancer prophylaxis”. Please remove and cite relevant reference.
  10. The section on endometrial cancer needs significant consolidation. Recent studies like those of O-Flynn et al have showed proof of principle that endometrial cancers can be detected in cervico-vaginal fluid based on cytology with sufficient diagnostic accuracy and a larger prospective study is currently underway (Jones et al BMJOpen 2021). Frais-Gomez and colleagues conducted an important systematic review and meta-analysis on the role of cytology in endometrial cancer detection. Proteins in urine, blood, and uterine aspirates have also been explored as potential biomarkers (See Njoku 2019). PAPSEEK and other genetic endometrial cancer studies are conspicuously absent.
  11. Although not evidence based, screening is often recommended in women at high risk of endometrial cancer such as Lynch syndrome and there are guidelines from the RCOG/ASCO/ESMO with regards to this. This was completely missed by the review.
  12. The concept of risk prediction in diagnostics was also missed out, with current trends aiming to including genomic /PRS scores in gynae cancer diagnostics.
  13. The authors should summarise the various sources of non-invasive sampling/liquid biopsy in gynaecological cancers such as vaginal tampons, urine, uterine lavage, blood etc, their pros and cons and where feasible, highlight studies where they have been explored as potential sources for cancer detection. (In a table)

Author Response

Dear Editor,

We would like to thank both Reviewers for their insightful and helpful comments and suggestions to improve our manuscript. Below, please find our detailed responses to each of them, highlighted in blue color.

Round 2

Reviewer 1 Report

The current version approves the manuscript

Author Response

Dear Editor

We would like to thank again to the Reviewer for insightful and helpful comments and suggestions to improve our manuscript. Below, please find our detailed responses to each of them, highlighted in blue color. All changes in the manuscript are trackable via the tracking changes tool.

  1. Line 410. The study cited is a narrative review, not a meta-analysis.

Corrected

  1. Table 1:on cervicovaginal secretions ref 178 is metabolites not proteins.

Corrected

  1. Table 3: I don’t agree that there are no prevention strategies for endometrial cancer. There are options like bariatric surgery-induced weight loss, progestins, prophylactic hysterectomy, etc depending on the risk factor for endometrial cancer.

Table 3 was supplemented

  1. Table 1: Cervicovaginal fluid should also have cytology as a method previously tried. Added into Table 1
  2. Line 652: Transvaginal ultrasound is more appropriate.

Corrected

  1. Several paragraphs need rewriting to enhance readability. Eg Lines 651-659

Rewritten

  1. Will benefit from extensive language editing

Language editing and proofreading were done by a native speaker.

Reviewer 2 Report

An important issue is, like highlighted by reviewer 1, the scope of the review which is quite broad and makes it difficult for any review group to effectively summarise the evidence base across the three cancer types.

Other Comments

  1. Line 410. The study cited is a narrative review, not a meta-analysis.
  2. Table 1:on cervicovaginal secretions ref 178 is metabolites not proteins.
  3. Table 3: I don’t agree that there are no prevention strategies for endometrial cancer. There are options like bariatric surgery induced weight loss, progestins, prophylactic hysterectomy etc depending on the risk factor for endometrial cancer.
  4. Table 1: Cervicovaginal fluid should also have cytology as a method previously tried.
  5. Line 652: Transvaginal ultrasound is more appropriate.
  6. Several paragraphs needs rewriting to enhance readability. Eg Lines 651-659
  7. Will benefit from extensive language editing 

Author Response

Dear Editor

We would like to thank again to the Reviewer for insightful and helpful comments and suggestions to improve our manuscript. Below, please find our detailed responses to each of them, highlighted in blue color. All changes in the manuscript are trackable via the tracking changes tool.

  1. Line 410. The study cited is a narrative review, not a meta-analysis.

Corrected

  1. Table 1:on cervicovaginal secretions ref 178 is metabolites, not proteins.

Corrected

  1. Table 3: I don’t agree that there are no prevention strategies for endometrial cancer. There are options like bariatric surgery-induced weight loss, progestins, prophylactic hysterectomy, etc depending on the risk factor for endometrial cancer.

Table 3 was supplemented

  1. Table 1: Cervicovaginal fluid should also have cytology as a method previously tried. Added into Table 1
  2. Line 652: Transvaginal ultrasound is more appropriate.

Corrected

  1. Several paragraphs need rewriting to enhance readability. Eg Lines 651-659

Rewritten

  1. Will benefit from extensive language editing

Language editing and proofreading were done by a native speaker.

Round 3

Reviewer 2 Report

The authors should be commended for persevering with the review process. The manuscript is very much improved.

Minor comment:

The appropriate ref for Line 415-the meta-analysis- is this:

Frias-Gomez, J. et al. Sensitivity of cervico-vaginal cytology in endometrial carcinoma: a systematic review and meta-analysis. Cancer Cytopathol. 128, 792–802 (2020)

Author Response

Dear reviewer,

Reference  "Frias-Gomez, et al. Cancer Cytopathol 2020128, 792-802" was added to the manuscript as you suggested.

Thank you very much.